TECHNICAL RELEASE

# epitopepredict: a tool for integrated MHC binding prediction

Damien Farrell[1],[*]

**1** UCD School of Veterinary Medicine, University College Dublin, Ireland

## ABSTRACT

A key step in the cellular adaptive immune response is the presentation of antigens to T cells. Computational prediction of T cell epitopes has many applications in vaccine design and immuno-diagnostics. This is the basis of immunoinformatics, which allows *in silico* screening of peptides before experiments are performed. With the availability of whole genomes for many microbial species it is now feasible to computationally screen whole proteomes for candidate peptides. epitopepredict is a programmatic framework and command line tool designed to aid this process. It provides access to multiple binding prediction algorithms under a single interface and scales for whole genomes using multiple target MHC alleles. A web interface is provided to assist visualization and filtering of the results. The software is freely available under an open-source license from https://github.com/dmnfarrell/epitopepredict

**Subjects**  Software and Workflows, Biomedical Science, Bioinformatics

## BACKGROUND

An essential step in provoking adaptive immunity, delivered by the activated CD8+ or CD4+ T cells, is the recognition of epitopes by T cell receptors (TCR). During this process, short peptides processed from self or foreign proteins may be presented on the surface of the cell and bound to major histocompatibility complex (MHC) proteins for binding to T cell receptors. Those peptide-MHC combinations that bind and activate an immune response are called epitopes. This is the major determinant step and is computationally predictable. The most effective approach is to estimate the binding affinity of a given peptide fragment to MHC class I or II molecules. Algorithms that can identify MHC-class I or MHC-class II binding peptides rapidly and accurately are essential for vaccine development, neo-epitope discovery, and immunogenicity screening of protein therapeutics. Many MHC binding prediction methods exist for both class I and II and have been comprehensively reviewed [1]. Currently the most effective methods are machine learning (ML) based approaches, which are trained on existing binding affinity data for a given MHC molecule. To do this, the peptide sequence is encoded and these features fit against the known affinity. To date, artificial neural networks (ANN) perform better at this task than other models such as linear regression. This is likely because the hidden layers in such networks are better able to account for the contribution of intrapeptide residue-residue interactions to the binding affinity. All methods vary in accuracy over MHC alleles depending on the availability of quality datasets. Pan-allele tools have been developed to deal with this issue [2]. These approaches can impute affinities for unknown alleles on the basis of neighboring MHC alleles with the highest sequence similarity and which have sufficient training data.

**Submitted:** 07 October 2020

\* E-mail: farrell.damien@gmail.com

Preprint submitted at https://doi.org/10.1101/2021.02.05.429892

By convention, peptides are selected using an arbitrary score threshold. For affinities, a threshold value of 500 nM is considered a binder and 50 nM a strong binder. The algorithms perform best at this classification task rather than re-producing exact affinities. This problem is intrinsic to ML-based approaches: the effect of the most dominant features is penalized intentionally to achieve better generalization on blind test data [3]. Another source of the inaccuracy is the loss of sensitivity of experimental assays at either very high or low binding affinity regimes. As a consequence, epitope candidates for subsequent experimental validation selected by ranking the affinities may not necessarily be the best approach. Percentage ranking is now often the recommended method [4]. However, the exact approach probably depends on the study in question. For example, searching a small number of proteins might mean taking the top ranked percentile from each sequence regardless of score. Threshold selection is discussed later in the examples.

## Strategies for epitope selection

A typical approach to binder selection is to select the top *n*th percentile per protein rather than using an absolute threshold value; however, for whole proteome studies, this is likely to introduce multiple false positives from peptides in proteins that would otherwise score very low globally. We therefore include in our method a global standardization of the score over the entire proteome, similar to that used by Bremel *et al.* [5] and others, by setting a global cut-off based on the top percentage of scores from the entire proteome. In addition, some alleles have a significantly higher score distribution and will dominate the results if a uniform score cut-off is applied; this applies in general to MHC binding predictors. Thus, separating global cut-off per allele so that low scoring alleles would be better represented is also advisable. This approach is consistent with recent work by Paul *et al.* [6] regarding allele-specific thresholds in MHC-I prediction. Three such alternative threshold strategies are provided in this library and discussed below.

## Binding promiscuity

Promiscuous MHC binders are defined in this context as those above the cutoffs in more than *a given number of* alleles. The rationale for this is that a peptide is more likely to be immunogenic in your target population if it is a binder in multiple alleles.

## Tools for epitope selection

Software for T cell vaccine development or neoepitope prediction currently concentrates on using the binding prediction or eluted ligand likelihood as the main selection methods. Typically, when a binding prediction tool is published, the authors will provide a binary that can be used on the command line or via a web interface. Some tools provide both. Command line tools offer better control and perhaps higher throughput but may be harder to use for a general user. Virtually, all of these tools require users to input each sequence and its allele separately. It is then difficult or impossible to integrate results from multiple sequences and alleles. The results are often in different formats and it is not possible to compare between algorithms, for example.

There are several computational pipelines that help a researcher to predict epitopes [7, 8]. EpiMatrix is a commercial desktop application designed for this purpose [9]. Commercial tools may be of high quality but are neither free nor open source, raising issues of reproducibility for academics. There is therefore a limited choice for users in readily available and easy to use tools.

## IMPLEMENTATION

This software is implemented entirely in Python [10]. To achieve some level of uniformity between prediction methods, a standardized programmatic interface for executing the binding prediction methods and processing the results was designed. The results from each method can then be processed and visualized in a consistent manner. Prediction methods are implemented by inheriting from a *Predictor* object. Each predictor may wrap methods from other Python packages or call command line predictors. For example the *TepitopePredictor* uses the epitopepredict.tepitope module provided with this package. This approach allows us to integrate a new prediction method in a relatively straightforward and consistent manner. The prediction methods always return a Pandas DataFrame (Pandas, RRID:SCR_018214) [11] in a standard format. The *predict_sequences* method is used for multiple protein sequences and can be run in parallel. This can take a GenBank or fasta file as input. For large numbers of sequences the prediction function should be called with save=True so that the results are saved as each protein is completed to avoid memory issues, since many alleles might be called for each protein. Results are saved with one file per protein/sequence in csv format. More details on how to use the Python API are given in the online documentation and in the example notebooks referencing the examples below.

The web application is implemented in Tornado [12] using the Bokeh [13] visualization library for making interactive plots.

## Supported MHC binding prediction tools

The following MHC binding prediction methods are supported through the API. This means they can be utilized via the command line tool. The first two are built into the package, the others require installation of external software by the user. NetMHC tools in particular have to be installed separately as they have a more restrictive academic license that does not allow them to be distributed by a third party or via a repository. Only the 'pan specific' versions of these tools are supported as they provide the best allelic coverage.

- TEPITOPEpan [14] is a position specific scoring matrix (PSSM) based algorithm. It uses 11 scoring matrices derived from combinatorial competitive binding assays on 11 HLA-DR alleles [15]. This method is pan specific and covers 700 HLA-DR molecules with unknown binding specificities based on pocket similarity to the original set of 11 library sequences. We have implemented this algorithm as a Python module, thus it comes with the package. It is fast but not as accurate in benchmarks as netMHCIIpan with fewer alleles covered.
- The BasicMHC1 predictor is a built-in MHC-I prediction method further detailed below. It is implemented using the scikit-learn [16] package. It only covers 103 MHC-I alleles and cannot currently be extrapolated for use with similar alleles (i.e. not pan specific) but provides a convenient alternative to the external tools.
- MHCflurry [17] is an MHC-I predictor also using ANNs trained on affinity measurements. It currently covers 112 human alleles. This is an open-source tool available via pip and thus easy to install. It is recommended for MHC-I predictions unless there are alleles not covered. The latest supported version is 2.0.1.
- NetMHCpan [18] is an artificial neural network algorithm covering many human and animal MHC-I alleles. This is trained on both MS eluted ligand data and binding affinity data. It therefore returns two properties: either the likelihood of a peptide becoming a natural ligand, or the predicted binding affinity. Version 4.1 is currently supported.

- NetMHCIIpan [19] is also an ANN, trained on binding data for multiple MHC-II alleles. Predictions are now extended to all HLA-DR, DQ and DP known sequences as from version 3.0 [20]. Both this tool and netMHCpan have the broadest species support of any algorithms. They both have good web interfaces but are covered by free non-commercial academic licenses and the local versions must be installed separately. Version 3.0 is supported.

## Available threshold methods

Thresholds for considering a peptide to be a binder are somewhat arbitrary. This tool provides three threshold methods. The results from each will overlap but will not be identical. These are applied per sequence/protein and per each allele using the currently loaded data. These three threshold methods are also available when calculating promiscuous binders. Ultimately, these are simply alternative methods of achieving the same result – reducing the set of predicted peptides.

**rank** – Selects the top ranking peptides in each sequence above a rank cutoff. This is the most frequently recommended method of binder selection in general.

**score** – Uses a single score cutoff for all peptides. Most binding predictors produce a binding affinity score (ic50) and a cutoff of 500 nM is common. There is no rule over which score cutoff is optimal, however. Some alleles will tend to produce higher scores. Also, unless some limit is placed on the number of peptides, large proteins will produce a lot of peptides compared to smaller sequences.

**global** – Allele specific 'global' cutoffs, this uses a percentile cutoff to select peptides using pre-calculated quantile scores for each allele. The global quantile scores were calculated for each prediction method using a set of sequences from known human antigens such as apical membrane antigen, Tetanus toxin, thrombopoietin, and interferon beta. Therefore, peptides can be selected as measured against a standard scale as opposed to their 'within protein' ranking. A typical value would be using the top 5% in each allele across all sequences. This technique is designed for selection of a small set of candidates from very large numbers of proteins, such as across a bacterial proteome. There is limited evidence to suggest that this selection method is superior to the other methods but we have used it for selectiing a small set of candidates from large numbers of proteins, detailed in example 2 below.

## A basic MHC-I predictor

This section details the built-in method for MHC-I binding prediction. It is implemented in Python using scikit-learn. The typical method of building such an algorithm is to encode the peptide amino acid sequences numerically in a manner that captures the properties important for binding. Then these features can be fit against their known binding affinities (or eluted ligand data) using a regression model of some kind. Several peptide encoding schemes were tested, including the NLF encoding scheme [21], OETMAP [22], a Blosum62 matrix, and a simple 'one hot' encoding method. One hot encoding was found to be adequate and the more complex schemes did not appear to offer any significant advantage. This may require further testing. For now, it is possible to create and train the predictor with any of these encoders. The regression model used is the *MLPRegressor* from sklearn, an implementation of a multilayer perceptron (MLP), a class of artificial neural networks. The data set used for training was primarily from the IEDB and was curated by the authors

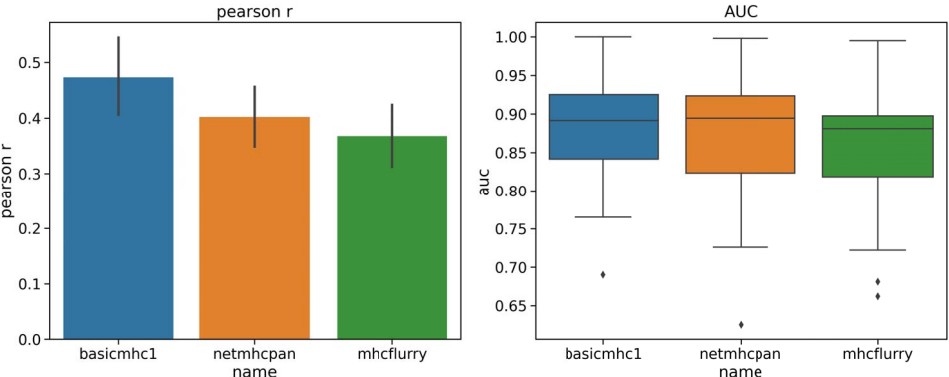

**Figure 1.** Performance of the basicmhc1 predictor compared to netMHCpan and MHCflurry for 40 human alleles. (a) Mean Pearson r and (b) mean AUC scores over all alleles. Only alleles with evaluation data for over more than 200 peptides were used. This test dataset used 9-mer peptides only.

of MHCflurry [17] from various sources. The regression model must be trained for each allele. When this is done, the model is persisted with the joblib module and can be re-loaded for new predictions for that allele. All of this functionality is encapsulated in the *BasicMHCIPredictor* class in epitopepredict. The predictor only supports 103 alleles currently and is not pan specific as of yet.

To test performance, a separate evaluation set of peptides originally created by Kim *et al.* [23] was downloaded from the IEDB. The training set sequences were subtracted from this set leaving 25,948 9-mer peptides. Only alleles for which there were more than 200 peptides were evaluated to give a reasonable performance estimate. This left 40 HLA alleles for testing. Both the Pearson correlation coefficient and the ROC AUC metric (with a threshold of below 500 nM set as a positive binder) were used as metrics. The results in Figure 1 show that our predictor performs similarly to the others with this test set. It is not meant to provide a definitive benchmark since these other tools have been more comprehensively benchmarked elsewhere. In particular, it can be hard to obtain a benchmark set of peptides that has not been used for training in one or more of the models.

In practical use, this predictor can be run directly from the API or command line without installing any other program. Models are trained once as needed for each allele/length combination using the current installed versions of scikit-learn and joblib. Once trained, each model is saved and can be re-used. Training only takes a matter of seconds for each model.

## RESULTS

In the following, we use several examples to illustrate the use of this package in practice with real data. These examples are available as Jupyter notebooks stored at https://github.com/dmnfarrell/epitopepredict/tree/master/examples. They are also archived permanently on Zenodo and the latest version is available there [24]. Some of these notebooks are also reproducible using the epitopepredict examples Code Ocean capsule (see Figure 2 [25]).

A

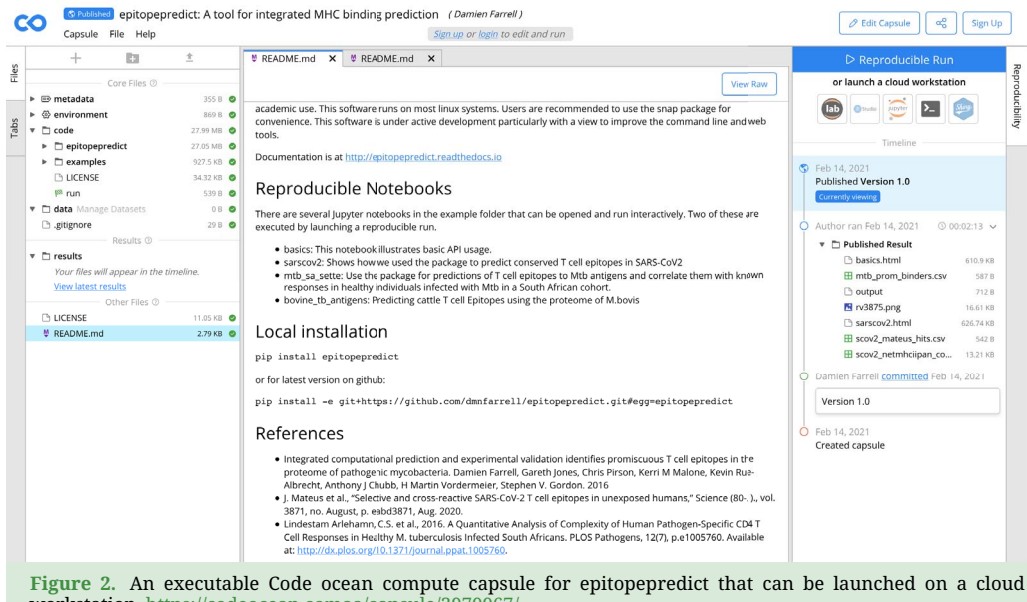

**Figure 2.** An executable Code ocean compute capsule for epitopepredict that can be launched on a cloud workstation. https://codeocean.comaa/capsule/3970067/

## Example 1: Predictions for selected antigens in *Mycobacterium Tuberculosis* – comparison with experimental data

A typical use of epitope prediction tools is to select a candidate list of peptides for testing from a large sequence space representing multiple potential antigens. This example provides a comparison of the three different selection methods in epitopepredict using a realistic example. It uses a set of known CD4 epitopes discovered in a study by measuring IFN-γ T cell responses to *M. tuberculosis* (Mtb) antigens in a healthy South African cohort [26]. The test data is available as supplementary tables in that paper. It comprises 75 15-mer epitopes selected from a set of known Mtb antigens.

Here, we performed a simple benchmark to find the percentage coverage of predicted MHC-II binders in two predictors, netMHCIIpan and Tepitope, using the three threshold methods for selecting promiscuous binders described above. These were then compared across a selection of cut-offs that each yielded a certain number of binders. Ideally we would want to produce as small a number of predicted binders as possible to reduce the number to be experimentally tested.

The sequences of all 29 proteins represented in the target set were retrieved and split into 15-mers. Then predictions were made for each of the 27 alleles in the target population tested in the study. This produced a list of 9299 peptides predicted for each allele. With epitopepredict, selection of promiscuous binders can be done easily with a single command. Binders promiscuous above thresholds in at least five alleles were selected.

The results are shown in Figure 3, with the plots showing the percentage of experimental peptides covered versus the number of predicted binders, corresponding to a certain cut-off in each method. It is seen that the 'rank' method is superior in both cases as it achieves a higher coverage with the lowest number of binders. All the curves level off at about 80% coverage. The 'rank' method may work better in this case partly because some of the epitopes were originally selected by prediction algorithms using a similar approach.

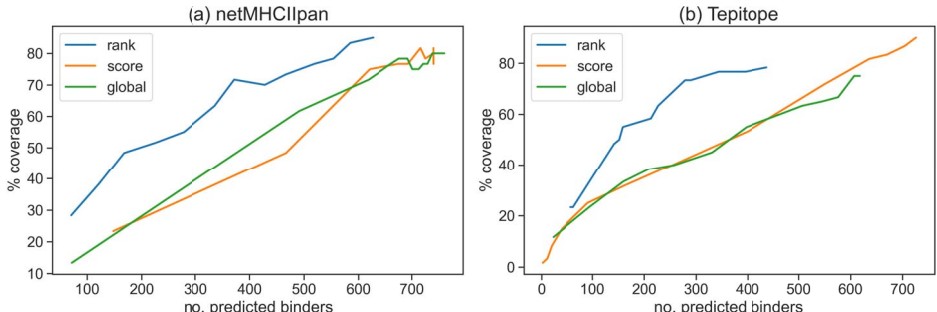

**Figure 3.** Performance of three binder selection methods showing the percentage coverage of experimental positive peptides by predicted binders at different cutoff levels. The higher the cutoff the more binders are predicted until the curves level off. Results are shown for (a) netMHCIIpan and (b) Tepitope.

## Example 2: Scanning the proteome of *Mycobacterium bovis* for CD4+ epitopes

We have previously used this package to prioritize CD4+ epitopes in the proteome of M. bovis (*Mycobacterium tuberculosis* variant bovis AF2122/97) for potential use in novel antigens for bovine tuberculosis [27]. The results are documented in the paper. Briefly, we performed binding predictions over the entire *M. bovis* proteome using two different binding predictors, netMHCIIpan [20], Tepitope [14]. For each set of results we found only promiscuous binders above an allele specific cutoff using the 'global' selection strategy. In addition, clusters of binders were detected to find areas of high binder density in each sequence. The assumption underlying this method is that ~20 mer peptides covering these regions will be more likely to yield at least one true positive epitope and hence elicit a T cell response. The results are a set of clusters for both prediction methods, ranked by number of binders per unit length. This has also been referred to as the 'epitope density' method [28]. We further contrasted this cluster selection with the more conventional ranking of top scoring binders. We also included random non-high-scoring peptides as a control. 20-mer peptides derived from these sets were synthesized and tested for IFN-$\gamma$ responses in *M. bovis* naturally infected cattle. Approximately 24% out of 270 peptides had high responses (using known epitopes as the baseline response). The random controls had no responses above this threshold.

This workflow was performed using an older version of this software. A newer and somewhat simplified form of the same analysis is now available as a notebook in the examples folder. Results from this output will be slightly different to our previous analysis since some of the extra steps have been removed, but the methodology is the same.

## Example 3: Predicting cross-reactive T cell epitopes in Sars-CoV-2

Eight months after the initial outbreak, puzzles remained about the human immune response to the SARS-CoV-2 virus. By then, a significant proportion in some large cities, such as New York, had been exposed. However antibody tests often revealed lower than expected rates of seropositivity in populations where the virus had spread [29]. It is almost certain that other components of the immune system were important in protecting individuals just as in other infectious diseases. Robust innate immune responses were one candidate. Another possibility is T cells. SARS-CoV-2 reactive CD4+ T cells had been reported



**Table 1.** Matches to the 10 cross reactive peptides found by Mateus *et al.* from our predicted binders shows hits in 6/10 cases.

| Sequence | Protein | Start | Hit from predicted set |
|---|---|---|---|
| PSGTWLTYTGAIKLD | N | 326 | GTWLTYTGAIKLDDK |
| SFIEDLLFNKVTLAD | S | 816 | FIEDLLFNKVTLADA, DLLFNKVTLADAGFI |
| YEQYIKWPWYIWLGF | S | 1206 | None |
| VLKKLKKSLNVAKSE | nsp8 | 3976 | VVLKKLKKSLNVAKS, EVVLKKLKKSLNVAK |
| KLLKSIAATRGATVV | nsp12 | 4966 | RQFHQKLLKSIAATR |
| EFYAYLRKHFSMMIL | nsp12 | 5136 | NEFYAYLRKHFSMMI, YLRKHFSMMILSDDA |
| LMIERFVSLAIDAYP | nsp12 | 5246 | None |
| TSHKLVLSVNPYVCN | nsp13 | 5361 | None |
| NVNRFNVAITRAKVG | nsp13 | 5881 | VNRFNVAITRAKVGI |

in unexposed individuals, suggesting pre-existing cross-reactive T cell memory in 20–50% of people [30]. It is possible that these were memory T cells generated from previous exposures to the human common cold coronaviruses (HCoVs), which circulate widely.

Mateus *et al.* [31] identified such cross-reactive CD4+ epitopes by generating 42 short term T cell lines specific to previously identified epitopes in PBMCs from unexposed donors. Then homologs to these peptides in the HCoVs were tested against these cell lines for a response. These tests were done in both unexposed and convalescent COVID19 patients. Cross reactivity was found in 10/42 of the T cell lines. Responding cells in unexposed donors were predominantly found in the effector memory CD4+ T cell population, though the consequences of this for protective immunity are not yet known.

Here we show how it's possible to predict such potential cross-reactive CD4+ epitopes just using the sequences.

The method used is as follows:

- Predict MHC-binders in each SARS-CoV-2 protein sequence and selected the top scoring candidates. Here, we use epitopepredict to predict the most promiscuous binders across the 8 most representative human MHC-II alleles. Each protein sequence is split into 15-mer peptides and scored.
- Select the top scoring peptides in each protein. In this case we select the peptides using the global cutoff method in the top 5% percentile for each allele. We also limit the total for each protein to 70 to prevent a very long protein like ORF1ab from dominating the selection.
- Calculate conservation of each peptide with it's closest homologous sequence in each of the other four HCoVs. Then rank them by percentage identity.

Using a limit of 70 peptides per protein, we found 282 predicted peptides. Out of these, 162 were conserved with >67% identity in at least one HCoV (most commonly with SARS-Cov-1). Note that for a peptide to be cross-reactive, it does not necessarily have to share all residues in common with its homolog. The 9-mer core binding sequence could be conserved with perhaps similar residues at the ends. We finally checked our 162 peptides against the 10 epitopes identified by Mateus *et al.* We found a hit in 6/10 cases, shown in Table 1. Some hits are two peptides overlapping in our set, which probably indicates the same core epitope.

## USAGE

### Command line interface

Installing the package provides a command line tool that is run from a terminal. It is envisaged that most users will utilize the package using this tool since it requires no programming knowledge. It provides pre-defined functionality with all inputs and settings specified in a text configuration file. One advantage of using configuration files is in avoiding long commands with multiple arguments that may be prone to causing errors. Also, configuration files can be kept to recall what setting was used for a particular workflow. Using this strategy, you can make MHC predictions with your chosen alleles and predictors in one run. If settings are left out generally defaults will be used so one can use a minimal file, simplifying usage. Other useful features of the tool are the ability to run predictions in parallel using multiple processing cores, the use of preset lists of alleles and resuming runs that have been interrupted without overwriting previous predictions. Results are saved to disk as text files and can be reread in a subsequent run of the tool without having to recalculate binding predictions.

By default, the command line tool will calculate the promiscuous binders to give you a unique list of peptides and include the number of alleles in which it is a binder. The table is ranked by this value and the maximum score over the alleles tested.

### API usage

A very basic example of how to use the library from the Python API is given here. More complex usage is detailed in the documentation.

```
import epitopepredict as ep

P = ep.get_predictor('basicmhc1')
from epitopepredict import peptutils

#get some random peptides, returns a list
seqs = peptutils.create_random_sequences(10)

#run predictions
res = P.predict_peptides(seqs, alleles='HLA-A*01:01')
```

**The above code returns a pandas DataFrame sorted by allele and rank.**

### Plotting

The API includes the ability to plot results for individual protein sequences for one or more predictor. In such plots, binders are shown as colored blocks at their position in the protein with multiple tracks, one per allele/method. This allows ready comparisons between methods. An example is shown in Figure 4. This shows binders for three MHC-class I predictors for an antigenic Mtb protein, Rv3875. Six HLA alleles are shown. We can see that each method has some overlap with the others.

### Testing

The command line tool can be tested by calling `epitopepredict -t`, which runs a set of sample Ebola virus sequences with the available prediction methods. Outputs are saved to a

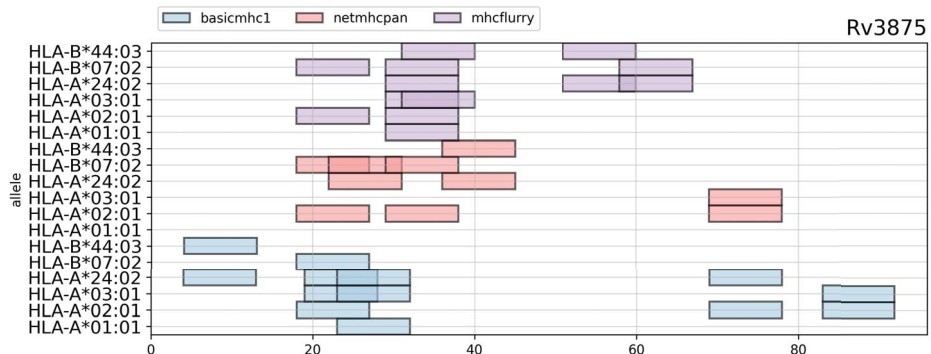

**Figure 4.** Predicted promiscuous binders in a sample sequence for three methods. Each method will have some overlapping peptides but they are usually likely to differ.

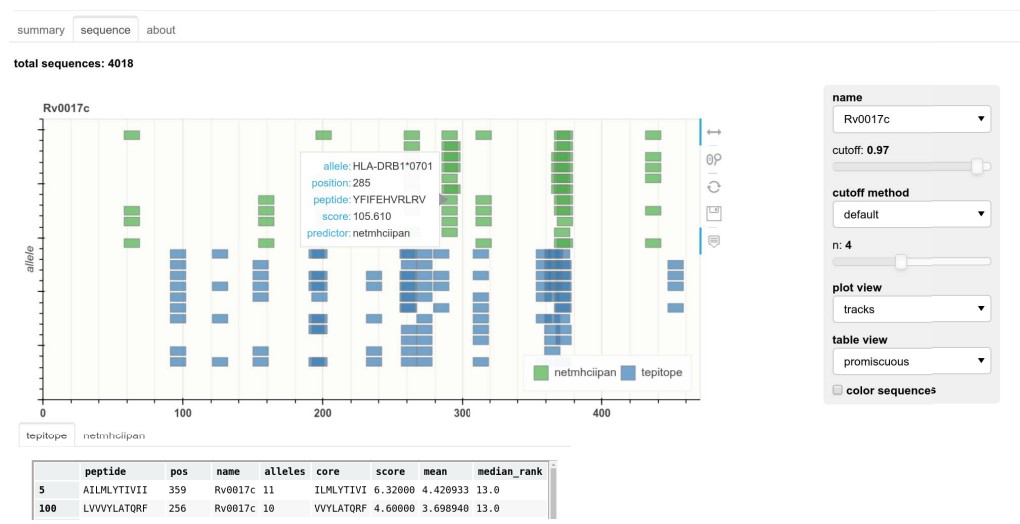

**Figure 5.** Web application showing results for a single protein sequence. Widgets can be used to select protein, cut-off levels and the type of plot.

folder called zaire_test. It should be noted that this is not used as a benchmark test since the algorithms used have all been tested independently. This is an example run for the user to check that the command line workflow is working and to inspect the outputs.

## Web application

A web interface that is launched from the command line can be used to view results from a set of predictions that have been previously made. This is an improved and much easier to use form of a previous web interface called epitopemap [32] and replaces it. Widgets can be used to select thresholds and the kind of plot shown. Currently two kinds of plots can be viewed, a sequence view and one that shows the peptides as colored blocks in tracks along the sequence, as shown in Figure 5. This web interface can be tested by running the test command above and then launching the web app using the zaire_test folder as input.

## CONCLUSIONS

This software provides a programmatic framework and command line interface for running multiple MHC binding prediction algorithms. This will be especially useful for performing high throughput calculations in many sequences and alleles. It is designed to scale for proteome scanning by allowing multiple processing threads to be used with any of the prediction methods. The API can also be easily applied to single sequences or small numbers of antigens. A web interface allows users to readily review results if they wish.

## AVAILABILITY AND REQUIREMENTS

Project name: epitopepredict
Project home page: https://github.com/dmnfarrell/epitopepredict
Archived version: v0.5.0 (DOI: 10.5281/zenodo.4056421)
SciCrunch Identifier: SCR_019221
Operating system(s): Linux, Unix
Programming language: Python
Other requirements: biopython, pandas, numpy, matplotlib, scikit-learn
Optional requirements: bokeh, panel (web app only) [33]
License: GNU General Public License v 3.0
Any restrictions to use by non-academics: None.

### Installation

This software should be run on a Linux operating system. Ubuntu is recommended but most major distributions will work well. Windows is not supported. If using Windows or macOS (OS X), users can simply install Linux using virtual machine software such as Oracle VM VirtualBox (https://www.virtualbox.org). Software is then installed using the online documentation. The installation process is very simple, requiring only a single typed command. Externally used MHC binding prediction algorithms do need to be installed separately, these are all freely available.

### Installing netMHCpan and netMHCIIpan

Due to license restrictions, these specific programs must be installed separately. They are free for academic users but require registration for the non-webserver version. You can go to https://services.healthtech.dtu.dk to fill in the forms that will give you access to the install file for the respective programs. The install instructions can then be found in the readme files when you untar the downloaded file, e.g. netMHCpan-4.1.readme. There are four steps detailed and the process is relatively simple. Remember to test that the software is working before you use it in epitopepredict.

## DATA AVAILABILITY

All computational work described here was implemented using Python. The code is provided as a Python package called *epitopepredict* under the GPLv3 license. Extensive use was made of the IPython (Jupyter) notebook environment [34] in prototyping the codebase.

Documentation for users is available at http://epitopepredict.readthedocs.io. Snapshots of the code are available in the *GigaScience* GigaDB respository [35], and a CodeOcean capsule is also available [25].



## EDITORS NOTE

This is the first *GigaByte* article to have an Executable Research Article (ERA) also available, which showcases executable versions of the figures. Utilising technology from Stencila, this allows interaction with the underlying code to produce programmatically-generated interactive versions of Figures 1, 3 and 4 [36]. Please click on the "View in Stencila" tab at the top of the paper to access this.

## FUNDING

This work was supported by the Irish Department of Agriculture Food and the Marine grant 15/S/651 (NEXUSMAP). DF was previously funded under an Irish Research Council Postdoctoral Fellowship (GOIPD/2015/475) for part of this work. The funders had no role in study design, data collection and analysis, decision to publish, or preparation of the manuscript.

## ACKNOWLEDGEMENTS

Thanks to Dr. Joseph Crispell for useful discussions on machine learning. Thanks also to Prof. Stephen Gordon for support during the development of this software.

## ABBREVIATIONS

ANN: artificial neural networks; hCoV: human common cold coronaviruses; MHC: major histocompatibility complex; ML: machine learning; MLP: multilayer perceptron; PSSM: position specific scoring matrix; TCR: T cell receptor.

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
