## [Reviewer Report]

Reviewer name and names of any other individual's who aided in reviewerYoonjoo ChoiDo you understand and agree to our policy of having open and named reviews, and having your review included with the published manuscript. (If no, please inform the editor that you cannot review this manuscript.)YesIs the language of sufficient quality?YesPlease add additional comments on language quality to clarify if neededThere are some minor typos. Perhaps this would not be a matter in other systems or viewer - all "fi" do not appear on my computer (Mac OS Preview), e.g. "affinity" -> "a inity", "artificial" -> "arti cial".Is there a clear statement of need explaining what problems the software is designed to solve and who the target audience is? YesAdditional CommentsThe purpose of this software is clearly stated and it will be very useful for researchers in relevant research fields.Is the source code available, and has an appropriate Open Source Initiative license <a href="https://opensource.org/licenses" target="_blank">(https://opensource.org/licenses)</a> been assigned to the code?YesAdditional CommentsAs Open Source Software are there guidelines on how to contribute, report issues or seek support on the code?YesAdditional CommentsIs the code executable?YesAdditional CommentsIs installation/deployment sufficiently outlined in the paper and documentation, and does it proceed as outlined?YesAdditional CommentsThe author recommended running this package on Linux machines, though it is written in Python. It would be great for a non-linux user to run TEPITOPE and BasicMHC1 (for a quick epitope screen). I pip-installed it on both Ubuntu and Mac OS (just to see whether I can run TEPITOPE and BasicMHC1). The installation on Ubuntu was very easy and running fine. The Mac OS installation failed, but perhaps not the trouble of epitopepredict (brew installed Python 3.9.0).Is the documentation provided clear and user friendly?YesAdditional CommentsIs there a clearly-stated list of dependencies, and is the core functionality of the software documented to a satisfactory level?YesAdditional CommentsHave any claims of performance been sufficiently tested and compared to other commonly-used packages? YesAdditional Comments(Definitely not mandatory at all but) It would be great this package also provides a wrapper for the IEDB tools.Are there (ideally real world) examples demonstrating use of the software? YesAdditional CommentsIs automated testing used or are there manual steps described so that the functionality of the software can be verified?YesAdditional CommentsAny Additional Overall Comments to the AuthorRecommendationMinor Revisions

---

## [Reviewer Report]

Reviewer name and names of any other individual's who aided in reviewerJayaraman ValadiDo you understand and agree to our policy of having open and named reviews, and having your review included with the published manuscript. (If no, please inform the editor that you cannot review this manuscript.)YesIs the language of sufficient quality?YesPlease add additional comments on language quality to clarify if neededThere are lot of spelling mistakes. Must be corrected before acceptance.Is there a clear statement of need explaining what problems the software is designed to solve and who the target audience is? YesAdditional CommentsThis is clearly explained In the manuscriptIs the source code available, and has an appropriate Open Source Initiative license <a href="https://opensource.org/licenses" target="_blank">(https://opensource.org/licenses)</a> been assigned to the code?YesAdditional CommentsThe source code is available on Github and it works as expectedAs Open Source Software are there guidelines on how to contribute, report issues or seek support on the code?YesAdditional CommentsIs the code executable?YesAdditional CommentsIs installation/deployment sufficiently outlined in the paper and documentation, and does it proceed as outlined?NoAdditional CommentsThe software depends on a number of external soft wares. Installation of the same need to be explained clearly in the manuscriptIs the documentation provided clear and user friendly?YesAdditional CommentsOverall the documentation is good. Doc-Strings need minor improvements to make it more comprehensive. Is there a clearly-stated list of dependencies, and is the core functionality of the software documented to a satisfactory level?YesAdditional CommentsThis is well explained in manuscriptHave any claims of performance been sufficiently tested and compared to other commonly-used packages? YesAdditional CommentsAdding a note on comparing the performance of different methods would be usefulAre there (ideally real world) examples demonstrating use of the software? YesAdditional CommentsIs automated testing used or are there manual steps described so that the functionality of the software can be verified?YesAdditional CommentsThe software is available on pip and Github. When the software is installed on pip,examples can not be run.Any Additional Overall Comments to the AuthorThe software developed is a python wrapper for a number of epitope prediction methods which are available. Unified architecture allows users to have easy access to all methods and compare the results of each method. Some of these methods/models have to be manually installed before the user can access it through the python wrapper. A new model trained by the authors has also been added additionally. users can utilize this prediction model without having to install any additional dependencies. 
Salient Features
The software also supports visual comparison of predictions
Users can select a target protein for epitope scanning 
users can prediction putative mhc1 and mhc2 epitopes using various predictive models using the python wrapper. 
Selection of best predictions possible
Visual comparison of predictions from different predictive models possible.

Highlights the positions of putative epitopes on the target protein sequence 


Overall the manuscript and software are quite comprehensive and can be accepted after minor revisions. RecommendationMinor Revisions